# Information Theoretic Study of COVID-19 Genome

**DOI:** 10.3390/e26030223

**Published:** 2024-03-01

**Authors:** Philippe Jacquet

**Affiliations:** Inria Saclay Ile-de-France, 91120 Palaiseau, France; philippe.jacquet@inria.fr

**Keywords:** genome, COVID-19, joint complexity, pattern matching

## Abstract

In this paper, we analyse the genome sequence of COVID-19 on a information point of view, and we compare that with past and present genomes. We use the powerful tool of joint complexity in order to quantify the similarities measured between the various potential parent genomes. The tool has a computing complexity of several orders of magnitude below the classic Smith–Waterman algorithm and would allow it to be used on a larger scale.

## 1. Introduction

The emergence of the pandemic disease, SARS-2 COVID-19, has been the major event of the last three years. There has been much speculation about the origin of the virus, and its future and past mutations. This is why the SARS-2 genome has attracted so much attention. The basis of information theory is to extract patterns and similarities between structures without necessarily relying on the functional meaning of common fragments, such as the meaning of words in texts or translated proteins in genomes. Nevertheless, we will show that the tools of information theory, such as joint complexity, are powerful enough to draw certain conclusions about recent speculations concerning the origin of the virus.

The article is organized as follows: first, we briefly introduce the concept of joint complexity and recall basic results on random sequences concerning “weak” matching, and establish some new results on “strong” matching. Next, we present our result on weak matching, which establishes that the COVID-19 virus is a descendant of a bat coronavirus. We also establish that the HIV virus should not be considered an ancestor, contrary to what some of the literature claims. Thirdly, we address the area of strong matching by analyzing similarities with recent bat coronaviruses.

The paper does not bring breathtaking new results in genetics, since most of the phylogenetic analysis on the COVID-19 genome have generated a huge amount of literature. However, most of these results have been obtained via methods that are very costly in computing power, for example, the classic Smith–Waterman algorithm [1]. The paper is more of an introduction to a much more powerful algorithm called joint complexity, which computes the alignments and similarity measure between two strings with a quasi linear processing cost, while the classic alignment algorithm is of quadratic cost. The new algorithm is expected to give performance as good as the BLAST algorithm’s performance [2]. However, the later algorithm is based on heuristic and experimental data, while the joint complexity algorithm is backed by information theory. This opens interesting new perspectives for the phylogenetic analysis by making affordable and rigorous segment insertion and deletion detection via joint complexity.

## 2. The Joint Complexity Tool and Performance

Let us take a finite alphabet A and a finite sequence *X* over A. A factor of *X* is a sequence v, which can be found in *X* without gaps or errors. In other words, there exists two other sequences, *u* and *w*, such that X=uvw. We call “string complexity” of *X*, C(X), the number of different factors of *X* [3,4]. Let *Y* be another sequence, we call “joint complexity”, J(X,Y) the number of different factors common to *X* and *Y*.

The joint complexity algorithm is a way to measure how similar two strings are, assuming that the higher the joint complexity is, the closer the sources of the strings are. Being general and non-alphabet dependent, it can be applied to natural language, genomes, and signal processing without specific learnings. For example, it has been successfully applied to Twitter monitoring, earning to the authors of Ref. [5] the third prize of SNOW-DC with an algorithm whose code did not exceed one page. The joint complexity algorithm has also been used as a fast and efficient ticket pre-classification engine in network management.

These quantities are easy to compute; indeed, the string complexity is simply the number of internal nodes in the extended suffix tree [6] (also called the spaghetti suffix tree). It can also be computed via the compressed suffix tree when the leaves point to a suffix in the string when this extension is unique. The compressed suffix tree of a string *X* occupies an average space of $\frac{\left|X\right|}{h}$, where *h* is the entropy rate of the text *X* and |X| its length. The natural way is to incrementally build the suffix tree like a regular tree [7], in this case, the average computation cost is $\left|X\right|\frac{\log|X|}{h}$ computation steps, each step being a symbol comparison and a pointer assignment or creation; however, the suffix tree can be built in a linear time thanks to the Ukkonen algorithm [8]. However, the Ukkonen algorithm might be inefficient when the string *X* is too large, since the tree traversal feature might generate many cache mismatches. In summary, one can evaluate the average cost of building the suffix tree, which should be between $\frac{\left|X\right|}{h}$ and $\left|X\right|\frac{\log|X|}{h}$ computing steps.

The genomes are written in the alphabet A={A,C,G,T}, made of the four nucleo-bases. Although the genomes sequences are not purely random, we will use randomly generated sequences over A for bench marking and comparisons. Since the bases mostly appear uniform in each genome, most of the time we will benchmark on memoryless uniform sequences on A. However all the results stated below have been obtained under more general sequence generation models, such as biased memoryless, Markov with finite memory, mixing models [9], etc.

**Theorem** **1**([10]). *The average complexity of a string X built on a memoryless or a Markov source satisfies:*
(1)$$\begin{array}{ccc} E[C(X)] & = & \frac{\left(|X|+1)|X\right|}{2}+\left|X\right|-\frac{\left|X|\log|X\right|}{h}-\left(\frac{1}{2}+\frac{\gamma }{h}\right)\left|X\right|\\ \; & \; & +O(\log|X|), \end{array}$$*where h is the per symbol entropy rate of the source model and γ is the Euler–Mascheroni constant.*

When the source model is uniform and memoryless on the four bases, we have h=log4. We notice that the string complexity in our models is quadratic, indicating that almost every factor comprised between any pair of positions in *X* is unique.

Computing the joint complexity of two strings *X* and *Y* consists of merging the common branches of their respective suffix trees and to enumerate their common internal nodes. If one of the common nodes is a leaf, then the exploration continues in the other tree by using the pointer contained in the leaf. The processing cost of the determination of the joint complexity is basically equal to the joint complexity when the latter is expressed in computation steps. To this cost, one must add the cost of building the suffix trees but the latter can be built separately and be re-used. The following theorem is trivial:

**Theorem** **2**.
*For two strings X and Y we have the inequality*

J(X,Y)≤min{C(X),C(Y)}.



### 2.1. Weak and Accidental Pattern Matching

By weak and accidental pattern matching, we mean the joint complexity between two random sequences *X* and *Y* when they have been independently generated.

**Theorem** **3**([6,7] Chapter 10). *When X and Y are of same length but generated on two different source models (e.g., a Markov transition matrix with different parameters): when |X|→∞*
(2)$$E\left[J\left(X,Y\right)\right]∼\frac{{\left|X\right|}^{\kappa }}{\sqrt{a\log|X|+b}}$$
*with κ<1, and some parameter a and b>0. When X and Y are of different length but on the same source model then, when both |X||Y| tend to infinity:*
(3)$$\begin{array}{ccc} E[J(X,Y)] & ∼ & \frac{\left(|X|+|Y|)\log(|X|+|Y|)-|X|\log|X|-|Y|\log|Y\right|}{h}. \end{array}$$

**Proof.** All the proofs are in Ref. [7], chapter 10, the major new result is in the refinement of the result about E[J(X,Y)], when *X* and *Y* are on the same source model but with different lengths. To simplify, we only hint the proof on a memoryless source. We know from Ref. [7] that J(X,Y)∼C(|X|,|Y|) where $C(z_{1},z_{2})$ is a solution to the functional equation:
(4)$$C\left(z_{1},z_{2}\right)=\left(1-e^{-z_{1}}\right)\left(1-e^{-z_{2}}\right)+\sum_{a\in A}C\left(p_{a}z_{1},p_{a}z_{2}\right)$$
with $p_{a}$, the probability of the occurrence of symbol *a* in a random sequence. If we denote $f_{\lambda }\left(z\right)=C\left(z,\lambda z\right)$, we get the following functional equation:
(5)$$f_{\lambda }\left(z\right)=\left(1-e^{-z}\right)\left(1-e^{-\lambda z}\right)+\sum_{a\in A}f_{\lambda }\left(p_{a}z\right)$$
whose asymptotic is obtained via the Mellin transform, as described in Ref. [11]. □

Since the logarithms appear in alternation in (Equation 3), one should not think that the expression of E[J(X,Y)] leads to large values. In fact, when |X|=|Y|, the asymptotic expression of E[J(X,Y)] is exactly equal to |X|, i.e., a quantity strictly linear in |X|. When |Y|≪|X| we get $E\left[J\left(X,Y\right)\right]∼\frac{\left|Y\right|}{2\log2}\left(\log\frac{\left|X\right|}{\left|Y\right|}+1\right)$.

The quantity J(X,Y) to which one must add the cost of building the suffix tree of *X* and *Y*, namely, $\frac{1}{h}\left(|X|\log|X\right|+\left|Y|\log|Y|\right)$ gives an estimate of the computing cost for the determination of the joint complexity, and clearly it is mostly linear in |X| and |Y| while the algorithm of Smith–Waterman is in |X|·|Y|. The processing cost is given in the computation step unit, which is a symbol comparison and a pointer assignment.

### 2.2. Strong Pattern Matching

We call strong pattern matching when the sequences *X* and *Y* are so close that they are just a slight alteration of each other. In this case, they are strongly dependent.

**Theorem** **4**.
*Let k≥1 be a fixed integer, assume X is generated by a memoryless or by a Markov source of finite memory, and Y differs via k symbol substitution. We have the estimate when |X|→∞:*

(6)
$$E\left[J\left(X,Y\right)\right]∼\frac{\left(|X|+1)(|X|+2-k\right)}{\left(k+2)(k+1\right)}.$$



Notice that when k=0, we find back the estimate E[C(X)] since C(X)=J(X,X) but only in the leading quadratic term in |X|, namely, ${\left|X\right|}^{2}/2$. In strong pattern matching mode the joint complexity remains quadratic.

**Proof.** To compute the leading term we look at the factors, which do not overlap the positions where the *k* substitution occurs between *X* and *Y*. These factors are common to both *X* and *Y*, and we know almost surely they are unique. Thus, our analysis rigorously is a lower bound, since we have no room to develop the upper bound proof. Let $J_{n}^{k}$ be the cumulated number of such factors considering all the nk combination of substituted positions between *X* and *Y*; therefore E[J(X,Y)]=Jnknk. We know that $J_{n}^{0}=\frac{(n+1)n}{2}$. The generating function $J^{0}\left(z\right)=\sum_{n}J_{n}^{0}z^{n}=\frac{z}{{\left(1-z\right)}^{3}}$ for |z|<1. We have the following recurrence:
(7)$$J_{n}^{k}=\sum_{m=0}^{n-k}J_{m}^{0}+J_{n-m-1}^{k-1}$$
which when translated in generating function, gives $J^{k}\left(z\right)=\frac{z^{k}}{1-z}J^{0}\left(z\right)+J^{k-1}\left(z\right)\frac{z}{1-z}$, which resolves in $J^{k}\left(z\right)=\frac{z^{k}}{{\left(1-z\right)}^{4}}\left(\frac{1+z}{{\left(1-z\right)}^{k-1}}-1+z\right)$. The asymptotic leading term is contained in $\frac{1+z}{{\left(1-z\right)}^{k+3}}$, which is $\sum_{n}\left(n+2-k\right)\frac{(n+1)n(n-1)\cdots n-k+1}{(k+2)!}z^{n}$. The coefficient of $z^{n}$ divided by nk gives the claimed asymptotic term. □

## 3. Accidental Pattern Matching on Genomes

The genome of COVID-19 totalises 29,866 bases (first variant discovered in 2020). In Figure 1, we show the joint complexity of the SARS-2 COVID-19 genome with the “Bat coronavirus HKU2” [12] (discovered in 2007), which has 27,165 bases. The SARS-2 genome is parsed from right to left, and the plot shows the joint complexity between this portion of the genome with the genome of the bat α. In dash, we show the average joint complexity between two random genomes obtained via the same uniform memoryless source over the four bases alphabet. This last plot is directly obtained via the formula (Equation 3). Since the last plot is below the joint complexity with the bat α, we can conclude that indeed the SARS-2 COVID-19 and bat α are related.

On Figure 2, we display the same plot but normalised with Formula (Equation 3). We add in red to the joint complexity with an HIV virus HIV-1 isolate 060SE from Sweden (1997) [13] (8732 bases) and see that the genomes are indeed unrelated. In fact, we obtained the surprising result that the plot is below the average value, which one would obtain from two random sequences indicating that some factors in SARS-2 COVID-19 and in HIV exclude each other.

However, in Ref. [14], the authors claim to have found 19 short portions of HIV genomes from different sources that appear in the SARS-2 genome. This paper was only a preprint, but it resulted in a lot of noise when it went public. Some found in its assertions the proof that the SARS-2 COVID-19 genome should have been forged for malignant purposes. Indeed, we have the following theorem:

**Theorem** **5**([7] Chapter 4). *Let $\left\{w_{1},w_{2},\ldots ,w_{k}\right\}$ a set of k different sequences. Let X be sequence built on a memoryless source. The probability that the sequence contains all the k factors together is smaller than ${\left|X\right|}^{k}P\left(w_{1}\right)\cdots P\left(w_{k}\right)$, where the $P(w_{i})$’s are the respective probability of occurrence of sequence $w_{i}$ from the memoryless source.*

The putative HIV fragments in SARS-2 genomes depicted in Ref. [14] each have an average length of 20 bases or more. Under the archetypal hypothesis that SARS-2 is typical of an uniform memoryless source for a statistical point of view, the probability to have all these 19 copied fragments in the SARS-2 genome would be $2\times 10^{-144}$. Thus, these accidental insertions would be virtually impossible.

Table 1 below lists the 19 matching genomes. Some must be reversed in order to obtain the claimed match.

Figure 3 shows the dispatching of the matching between SARS-2 genome [12] and the 19 HIV genomes (plus the bat coronavirus alpha, which is number 20). The figure has been created the following way: The SARS-2 genome has been cut in slices of length 24 bases starting every 2 bases. For each HIV candidate genome *X*, we compute its joint complexity with every slice $Y_{1},\ldots ,Y_{14,933}$ of the SARS-2 genome, the processing cost for each slice is approximately $\frac{24}{\log4}\left(\log\frac{\left|X\right|}{24}+1\right)$ computation steps according to Theorem 3. The total processing cost for the whole operation is approximately $29\times 10^{6}$ computation steps since h=log4, including the computation of the suffix trees. With the algorithm of Smith–Waterman, it would have taken $2.8\times 10^{9}$ computation steps.

All the collected results give a mean and a variance, then we display the deviation from the mean in multiples of the standard deviation for each slice, knowing that we can obtain negative values. This way the accidental matching will be made more apparent. The blue vertical lines are the positions where the maximum deviation appears for each HIV genome. For example, for the genome 18, the position of the maximum is 20,400 and has an intensity 15 times the standard deviation, which is very large. The brown plot gives the maximum of the deviations obtained with the joint complexity algorithm over the 20 genomes for each slice of the SARS-2 genome. Notice that second largest deviation is obtained with ”Bat coronavirus HKU2” indicated by index 20. Fifteen times the standard deviation would mean a probability around $7.2\times 10^{-100}$ in a pure Gaussian context. It should be noted that the high peaks correspond to the slices with almost an exact copy in the other genome, while the weaker peaks are when there are more mismatches as an illustration of the strong matching theory. The paper [14] lists sequence matching up to three or four mismatches.

As a matter of comparison, we display in Figure 4 the same plot but with the reversed SARS-2 genome. The maxima are way less dramatic. However, we notice that all these genomes are coming from very diverse sequences, on HIV-1, other on HIV-2, and some on ape origin (the simian IV). Many have been even tested with a reversed sequence. We can imagine the authors may have tested much more sequences than the 19 selected sequences; there may be an explanation of this paradox here. Let *M* be the cumulated number of bases of the tested database. Due to the large sampling of HIV and HIV-related sequences in the databases, we can estimate *M* to the order of half a million bases. Processing the joint complexity of the concatenation of these genomes with the slices of the SARS-2 genome would take only $8.4\times 10^{6}$ the computation steps according to Theorem 3. This is an estimation because we did not actually perform the global search. However, there is a surprising estimated reduction in the complexity of the global search compared to the previous individual searches. It comes from the fact that we would have built a single suffix tree for the concatenated genome and make a single pass into this unique suffix tree for each slice instead of doing 19 searches in 19 suffix trees to detect the strongest matcher. The number of positions that can be tested for each match is M|X|, with *X* being the SARS-2 genome sequence. If 20 is the size of expected matches, the average number of matches of length 20, is $M|X|4^{-20}$ in the uniform memoryless model. If we include the possibility of up to three errors in the matching, we have to multiply this number by 203. Using Tchebychev inequality, the probability to have *k* matches is smaller than the average number of matches M|X|4−20203 divided by *k*, with which we would obtain the following:(8)P(19matches)≤M|X|4−2019203∼0.8.

Considering the reversed genome would simply multiply this figure by two. Clearly the matches are no longer exceptional; however, one could argue that the Tchebychev upper bound is very rough and the real probability could be smaller. However, it should be noted that the probability becomes much larger when the data are strongly positively correlated. This is confirmed by Figure 5, which displays the numerous accidental matching between HIV-2UC1 and the other matcher genomes. It should be stressed again that with the samples dating from around 1993, the DNA editing technology did not exist.

## 4. Strong Pattern Matching on COVID-19 Genomes

In this section, we try to analyse the hypotheses of the relation of COVID-19 with its potential ancestors and descendants. The currently accepted family tree is summarized in the following Figure 6:

In short, the first putative ancestor is the bat coronavirus “Bat coronavirus HKU2” [15] (we have already called it bat-α), which was discovered in 2007 and has 27,165 bases. The next ancestor is the bat coronavirus RaTG13 [16,17], which was discovered in 2013 and has 29,855 bases (let us call it bat-β). Then the first SARS-2 COVID-19 coronavirus for humans, discovered in late 2019, and another bat coronavirus RaCCS203 [18], discovered in early 2020 and has 29,775 bases; thus, the pivot genome is the bat-β RaTG13. In the following figure, we sliced the bat-β genome in slices of 50 bases each and computed the joint complexity with other genomes. Figure 7 shows the bat-β’s slice joint complexity with the whole genome bat-α. Apparently, the ancestor seems too distant to look nothing more than random, we are on a weak pattern matching level indicated by the lower dashed horizontal line determined by the estimate produced in (Equation 3). Figure 8 shows the bat-β slice joint complexity with its whole genome. In this case, the joint complexity naturally finds the position of the slice in the whole genome as its best match and the figure basically shows the complexity of the slice and illustrates the formula of Theorem 1 (indicated by the dashed upper line). Between the two lines lies the transition between weak pattern matching and strong pattern matching.

Figure 9 shows the bat-β’s slice joint complexity with its whole SARS-2 COVID-19 genome. Surprisingly, the slice joint complexity seems to be in a strong matching regime (very close to the upper horizontal dashed line), indicating a high degree of similarity. This is unexpected because there is the same time span between the discovery of bat-α and the discovery of bat-β than there is between the discovery of bat-β and the SARS-2 COVID-19 (6 years in both cases). Even more surprising as shown in Figure 10, is there is even more similarities with SARS-2 than with the genome of the last bat coronavirus RaCCS203, although the latter is for the same specie (bat), and the former is for two different species (human versus bats). Indeed, the plot of pattern matching between bat-β and RaCCS203 shows many places where the pattern matching is weak, in particular between the position 21,500 and 24,000 probably indicating the possibility of a large insertion of exogen genetic material.

Since we are in the context of strong pattern matching, the processing cost is larger than with the accidental pattern matching. If we use the Theorem 2, we use the fact that when |Y|≪|X|$J\left(X,Y\right)\leq C\left(Y\right)\leq \frac{\left|Y|(|Y|+1\right)}{2}$ we obtain an upper bound of $91\times 10^{6}$ computation steps.

The three genomes are so close that we can make correspondence with the segments of each genome with the segment in the other genome. Via a straightforward adaptation of the joint complexity program, we can compute the offset between the segments in one genome with the segments in the other genome. It consists of spotting the largest common factor instead of enumerating the common factors. In terms of programming, it is just replacing the operator of the summation evaluation with the operator of the maximum evaluation. Thus, for each slice of bat-β, we detect the position of its largest match in the other genome. The difference in positions between the two matches in their respective genome is the offset. If the offset is positive, then the match is in advance in the first genome compared with the second genome; otherwise, when it is negative it is in advance in the second genome.

Figure 11 shows the offset per bat-β slice with the SARS-2. There are the following two surprises: firstly, the surprise that except for an extreme minority of slices marked by the three dotted vertical blue lines, the offset is constant and flat and increases from −26 to −16. The offset stability indicates that the mutation sequence between the two genomes are mostly substitution. The three slices, which do not fit well, are slices where the substituted bases are too numerous and corrupt the largest match to make it jump by a large value, since the correspondence can be anywhere in the genome sequence, such as in the interval [−29,855,+29,855]. For the readability of the figure, we have truncated the abscissas. The second surprise is that the offset monotonically increases, indicating that the mutation happened via insertions and never by deletion. That is against the common belief that virus mutations mostly proceed by deletion. Maybe it is the consequence of the inter-species transfer from bat-β to SARS-2.

Figure 12 shows the result of the same exercise of offset determination from the bat-β genome to the last bat coronavirus RaCCS203. Contrary to the transition between the bat-β genome to the SARS-2 genome, the offset value is decreasing, sometimes sharply, indicating that the mutation is proceeded more by deletion than by insertion, confirming the natural trends in virus evolution. However, we notice some small insertions at some positions where the offset value slightly bumps up. We again notice the large corrupted area between position 21,500 and position 24,000. However, the offset value drops after too; thus, this exogen insertion plus the following deletion finally does not push the material to the right.

Figure 13 displays the mismatch rate between each slice of the bat-β genome and its corresponding slice in the SARS-2 genome in blue. We see very few strongly corrupted slices with poor correspondence, while elsewhere the substitution rate oscillates between 0 and 10% per 50 base slices. In green, we do the same exercise with the RaCCS203 genome. Although in the same lineage, the corruption are much more important. We again notice the area between 21,500 and 24,000 where the ratio Hamming distance to the length is around 65%, 10 points below the expected 75% if both portions were uniformly and independently generated, but which can be explained by the fact that the largest match should at least be around 5–6 bases.

## 5. Conclusions

We have presented an analysis of the COVID-19 genome and about its possible origins via the pure information theoretic tool. Our investigations do not address any medical and biogenetic considerations, and are mainly based on the pure randomness in the genome mutation process; therefore, they cannot lead to definite answers. Anyhow, we can establish that the accidental insertion of HIV segment in the SARS-2 COVID-19 is not so exceptional and can be easily explained by the abundance of existing materials in the genetic database of HIV. On the other side, the strong pattern matching with the putative ancestor bat coronavirus RATG13 is a surprise since the two sequences are more than 6 years apart and attached to two different species. The matches are much weaker with the putative descendants of RATG13 and RACS203, despite the fact that they are both related to bats.

However, beyond any phylogenetic conclusions, which lay beyond the scope of this work, this paper is an opportunity to advertise the formidable efficiency of the joint complexity tool to capture similarities in sequences. It provides both accuracy and cost-saving methods by being quasi linear in complexity. 

## Figures and Tables

**Figure 1 entropy-26-00223-f001:**
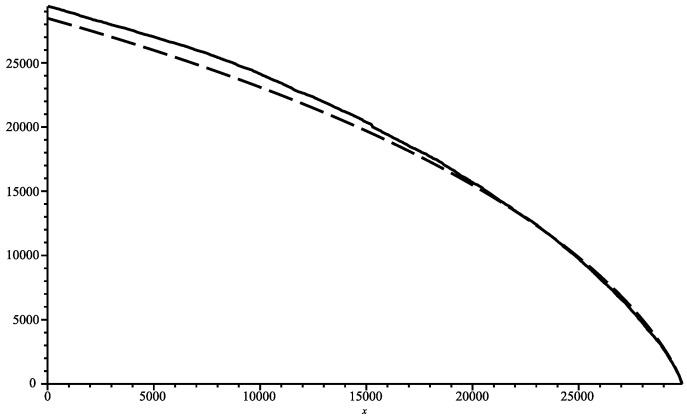
Joint complexity of SARS-2 genome with bat coronavirus alpha (solid), with random genomes (dashed).

**Figure 2 entropy-26-00223-f002:**
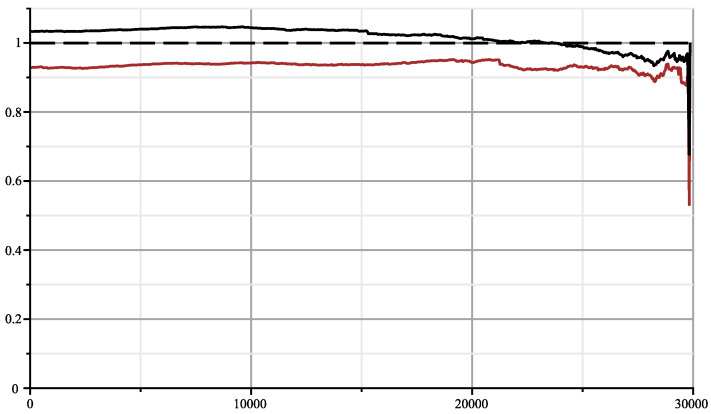
Normalised joint complexity of SARS-2 genome with bat-α, and with HIV genome (red).

**Figure 3 entropy-26-00223-f003:**
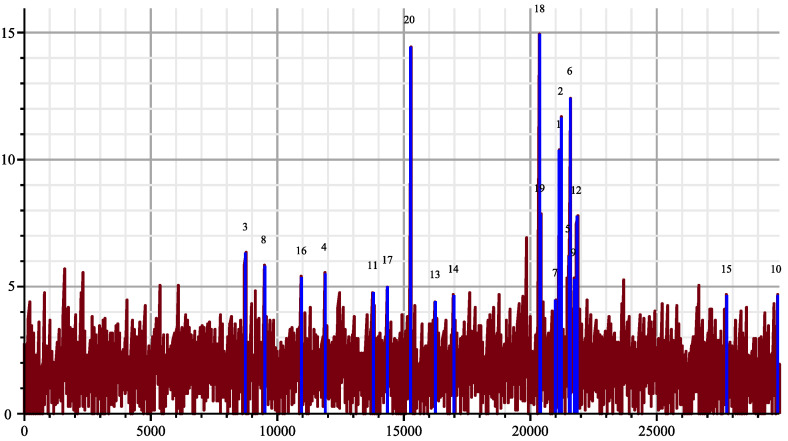
Joint complexity deviations of SARS-2 genome with the 19 HIV genomes.

**Figure 4 entropy-26-00223-f004:**
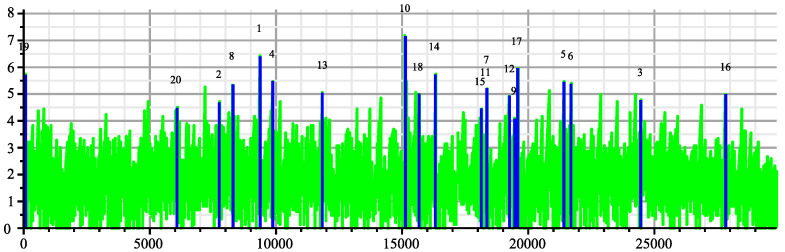
Joint complexity deviations of the Reverse SARS-2 genome with the 19 HIV genomes.

**Figure 5 entropy-26-00223-f005:**
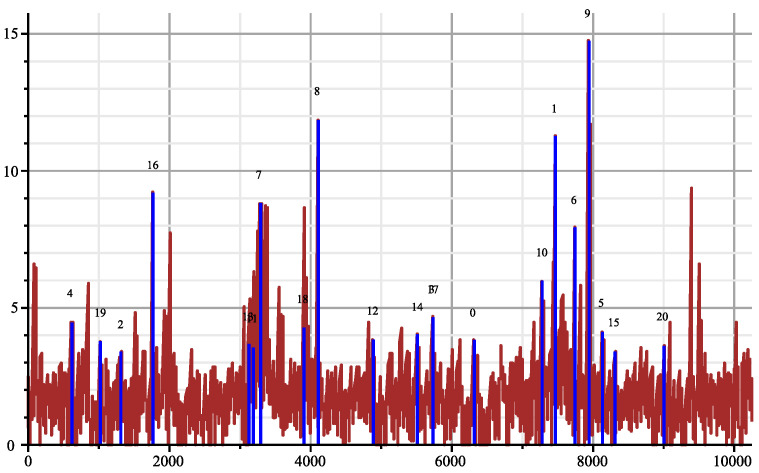
Joint complexity deviations of HIV-2UC1 genome with other matchers.

**Figure 6 entropy-26-00223-f006:**
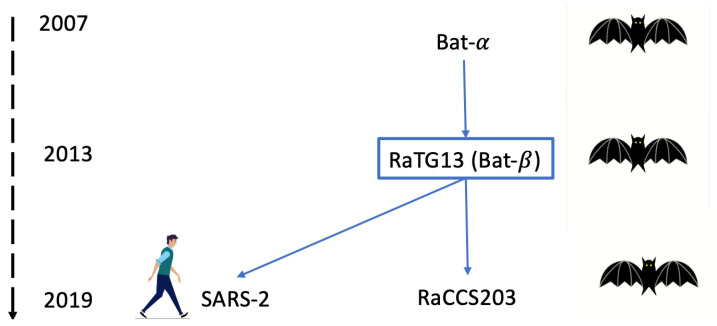
Putative genealogical tree of SARS-2 COVID-19.

**Figure 7 entropy-26-00223-f007:**
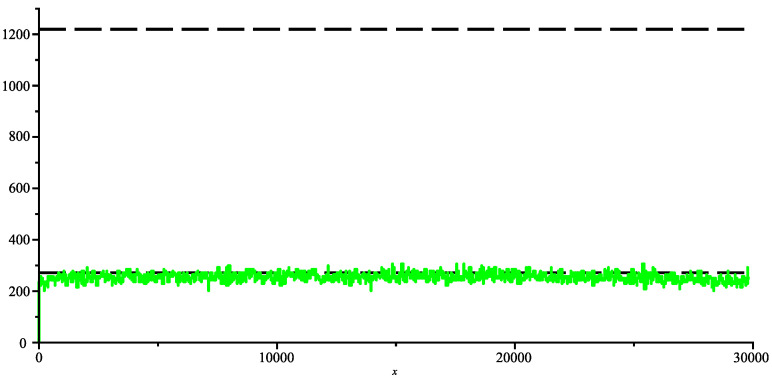
Joint complexity of bat-β genome with bat-α.

**Figure 8 entropy-26-00223-f008:**
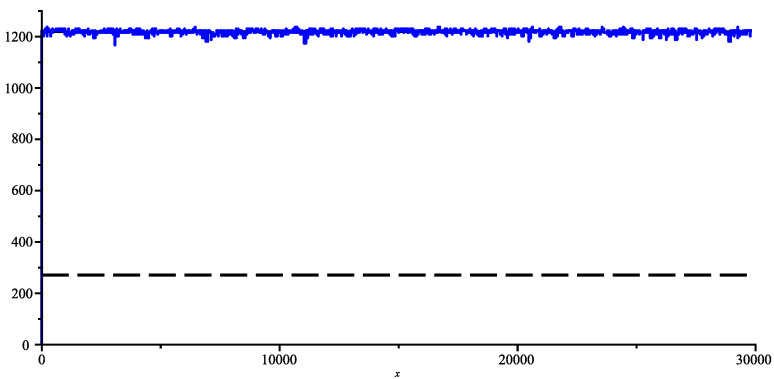
Joint complexity of bat-β with itself.

**Figure 9 entropy-26-00223-f009:**
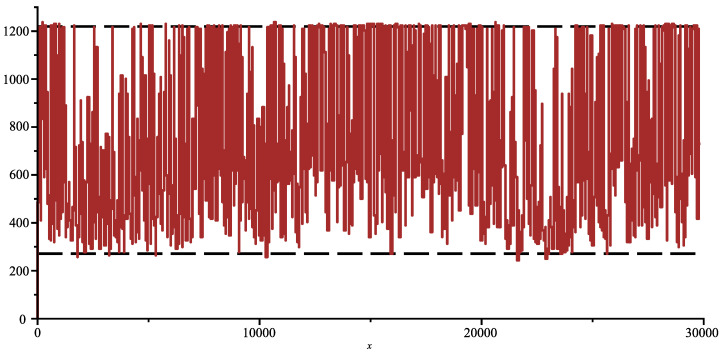
Joint complexity of bat-β genome with SARS-2 COVID-19 genome.

**Figure 10 entropy-26-00223-f010:**
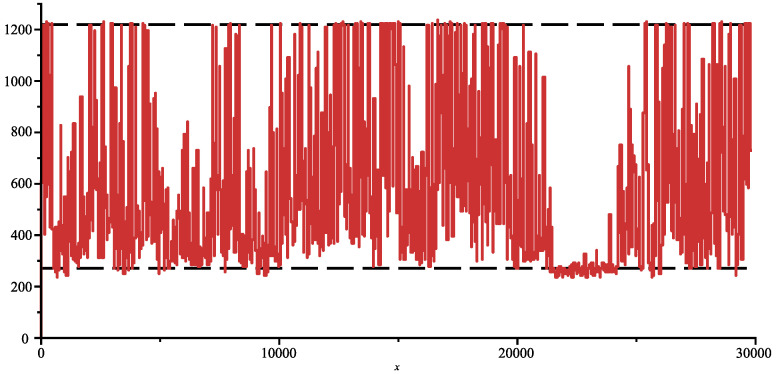
Joint complexity of bat-β with the RaCCS203 genome.

**Figure 11 entropy-26-00223-f011:**
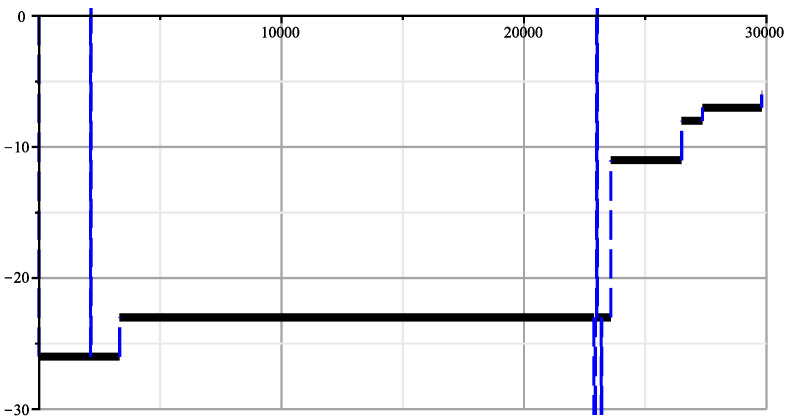
Local offset value of the bat-β genome with SARS-2 COVID-19 genome.

**Figure 12 entropy-26-00223-f012:**
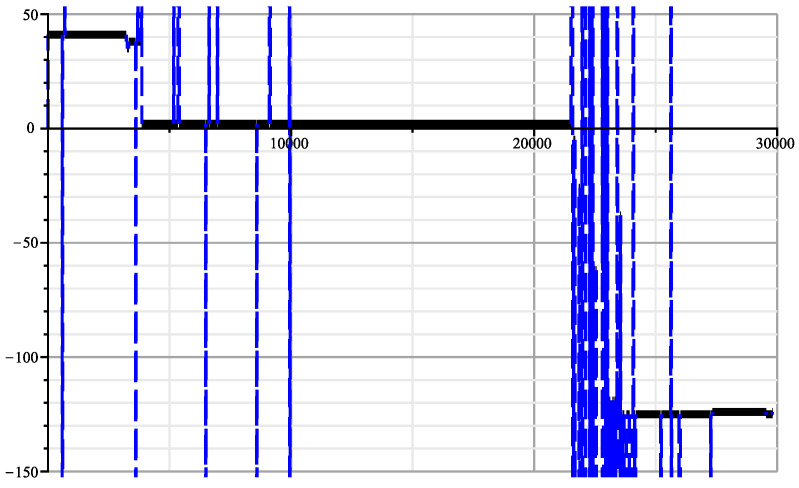
Local offset value of bat-β with with the RaCCS203 genome.

**Figure 13 entropy-26-00223-f013:**
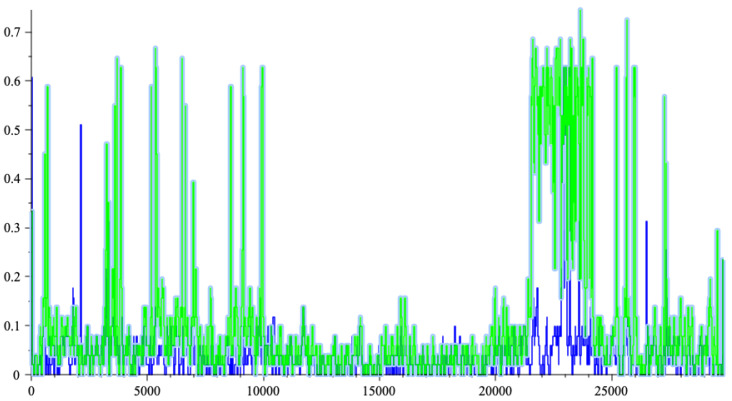
Mismatch rates. between the bat-β genome slices, corresponding SARS-2 genome slices (blue), and corresponding RaCCS203 genome slices (green).

**Table 1 entropy-26-00223-t001:** The 19 matching genomes origins.

Index	Length	Genome Origin	Index	Length	Genome Origin
1	236	HIV2-56-Isolate	11	10,401	HIV2-UC1
2	8840	HIV1-060SE-Sweden *	12	993	HIV2-Senegal *
3	2053	HIV2-Bissau *	13	2604	HIV1-Malawi *
4	9167	Simian-VSAA2001 *	14	2612	HIV1-Russia *
5	607	HIV1-clone-ML1592 *	15	3149	Simian-CM545 *
6	344	HIV2-Verde	16	9744	Simian-KM378564
7	920	HIV2-106	17	704	HIV1-EU184986 *
8	10,018	Simian-TAN5	18	125	HIV1-AY516986
9	1100	Simian-P18	19	2630	HIV1-HQ217329 *
10	1157	HIV1-19828	20	27,510	Bat-coronavirus-HKU2

* Genome is inversed for the matching.

## Data Availability

The data presented in this study are available on request from the corresponding author.

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
