# Peer review of "Information Theoretic Study of COVID-19 Genome"

_entropy, 2024, doi:10.3390/e26030223_

Round 1

Reviewer 1 Report

Comments and Suggestions for Authors

The author presents a study of COVID19 and attempts to identify the potential origin of COVID19 virus. I have the following comments. 

1. While the overall topic is interesting, the topic has been in-depth analyzed previously many times. How does this work differ from better from previous similar work?

2. I suggest the author try to explain the information theory in a more lament language in the context of COVID19.

3. Why is the method chosen better than BLAST or Smith-Waterman alignment?

Author Response

Thank you for the careful review.

I have extended the information theoretic part by giving more insights about the performance of the joint complexity computation via suffix trees. The joint complexity processing is linear in the size of the database and is thus of much lower complexity than Smith-Waterman algorithm, the latter being quadratic.   I have not drawn precise comparisons with BLAST algorithm because the latter is difficult to analyze since it involves a learning phase based on various heuristics. But I expect the processing cost at most similar with the difference that joint complexity is backed by information theory.

I have inserted additional references accordingly.

Reviewer 2 Report

Comments and Suggestions for Authors

This study presents an elegant analysis of Covid 19 genome's relationships with other relevant genomes from an "agnostic" information-theoretic perspective. The analysis is carried out via the relatively novel concepts of joint complexity and weak/strong matching. The analysis is rigorous and convincing. 

My main concern with the study is two-fold: first, it reaches the conclusions that are not particularly controversial, yet the author positions them as something of a revelation. Second, and related --- the author does not present a comprehensive picture of the complementary analyses in the literature (phylogenetics, etc.). In general, it is difficult to understand how the author's analysis and results fit in the broader picture --- there are hundreds of papers on Covid 19 phylogenies alone, and yet the References section in the manuscript is downright anemic. 

In this reviewer's opinion, the manuscript would benefit greatly from the much expanded Background and Conclusion sections. Taking an agnostic position ("we are not interested in the bio/medical aspects and clues, it's just pure information theory!") might be refreshing and potentially productive, but the results should be assessed in the general context to be of interest to the broad audience. 

Minor: Reference missing in line 38

Author Response

Thank you for your careful review,

I acknowledge that the paper does not provide breaking new insights in the SARS-2 genetic analysis. My point was more on the use of pure information theoretical tools for the analysis of the genome. I have put more emphasis on the main novelty of the paper which is the application of the joint complexity as a fast and efficient tool to detect similarities and pattern matchings without learning phases and heuristics. In this revised version I have amplified the computational aspects of the joint complexity processing and tried to compare in theory and practice with the classic alignment tools.

More references are added.

Round 2

Reviewer 1 Report

Comments and Suggestions for Authors

The authors addressed my comments.

Reviewer 2 Report

Comments and Suggestions for Authors

A moderately expanded context (including in Abstract and Introduction), absence of which was my major concern in the first review round, should be helpful in attracting the proper audience for this study. The paper is ready for publication now.